# Effect of a High Protein Diet at Breakfast on Postprandial Glucose Level at Dinner Time in Healthy Adults

**DOI:** 10.3390/nu15010085

**Published:** 2022-12-24

**Authors:** Keyi Xiao, Akiko Furutani, Hiroyuki Sasaki, Masaki Takahashi, Shigenobu Shibata

**Affiliations:** 1Laboratory of Physiology and Pharmacology, School of Advanced Science and Engineering, Waseda University, Shinjuku-ku, Tokyo 162-8480, Japan; 2Faculty of Home Economics, Aikoku Gakuen Junior College, Edogawa-ku, Tokyo 133-8585, Japan; 3Institute for Liberal Arts, Tokyo Institute of Technology, Meguro-ku, Tokyo 152-8550, Japan

**Keywords:** protein, breakfast, skipping lunch, second meal, postprandial glucose

## Abstract

This study aimed to examine the effect of high protein breakfast diet with or without lunch on the postprandial glucose level during the day. A randomized, crossover design that recruited 12 healthy young participants (three men and nine women) was performed and four trials (normal breakfast + skipped lunch, high protein breakfast + skipped lunch, normal breakfast + lunch, and high protein breakfast + lunch) were conducted in two weeks. During each trial, breakfast, lunch, and dinner on the trial day, and dinner before the trial day, were provided as test meals, and the meal timing was fixed. Continuous glucose monitoring (CGM) was used to assess the blood glucose level during the whole experiment. Incremental area under the curve (iAUC) of the postprandial glucose level was calculated. The results suggested that compared with normal breakfast, high protein breakfast suppressed the 3 h iAUC of postprandial glucose level after breakfast (*p* < 0.05 or *p* < 0.0001) and 1.5 h iAUC after lunch (*p* < 0.01). During lunch, high protein breakfast diet suppressed the dinner and overall day postprandial glucose level (*p* < 0.05 vs. normal breakfast), but no significant difference was observed when skipping lunch. Our findings indicate that high protein breakfast could suppress the breakfast postprandial glucose level, as well as following lunch and dinner, but this effect on dinner was attenuated when skipping lunch.

## 1. Introduction

Postprandial hyperglycemia may increase the risk of many lifestyle-related diseases like cardiovascular diseases [1], hypertension [2], and obesity [3], and is the major cause of death in type 2 diabetes (T2D) [4]. Therefore, identifying ways of reducing postprandial glucose are essential for preventing these lifestyle-related diseases.

Recently, second meal effect, which is defined as the influence of food intake in one meal on the postprandial glucose level after the second meal, has become a popular strategy for controlling the postprandial glucose level [5]. Glycemic index (GI) is usually used to describe the rate at which glucose level increases after eating a kind of food. According to the second meal effect, consumption of a lower GI food has been proved to not only suppress the postprandial glucose level after that meal but also after the second meal. In healthy adults, eating biscuits with mulberry or barley leaf beverage as afternoon snacks prevents a dinner-induced high blood glucose level increase [6]. Compared with normal biscuits, biscuits with high dietary fiber between lunch and dinner suppresses the postprandial blood glucose level after dinner and even after the next day breakfast [7].

The balance of the three necessary macro nutrients, protein, fat, and carbohydrate (PFC), plays a key role in healthy daily life. According to the Overview of Dietary Reference Intakes for Japanese [8], the ideal PFC balance for those aged 18 to 49 years regardless of gender, is from 13~20% protein, 20~30% fat, and 50~65% carbohydrate. However, for diabetes prevention and treatment, recent studies are focused on adjusting the balance to make it more suitable to control the blood glucose level. High fat meal has been shown to lead to higher postprandial glucose level following subsequent meals than high carbohydrate meal [9]. However, with protein, the second meal effect usually suppresses the postprandial glucose level [10,11]. A previous study showed that high protein lunch could suppress the dinner postprandial glucose level [12]. Therefore, when thinking about changing the PFC balance to control the postprandial glucose level, raising the protein ratio may be a better choice. Considering the potential consequences of protein deficiency or excess, 10~35% of caloric intake is recommend as the US acceptable macronutrient distribution range (AMDR) for protein [13]. Therefore, we set the highest protein percentage of the daily test meal to 32% in this study.

In modern society, heavy workload with social stress and shift work lead to increasingly irregular lifestyle and meal skipping, in both adults [14] and children [15]. Sometimes, even when food is available, many people tend to skip meals to conserve time or just because they find it difficult to leave the work desk [16]. The importance of breakfast is widely acknowledged, and skipping of breakfast is related to higher risk of chronic diseases [17], obesity [18], T2D [19] and cardiovascular diseases [20]; however, few studies have focused on the skipping of lunch [21]. Furthermore, during the coronavirus disease 2019 (COVID-19) pandemic, people spent more time at home and the daily habits have changed considerably. Thus, along with the habit of remote work, more people began to eat breakfast, which they usually skipped because of work before the pandemic; however, the ratio of skipped lunch increased [22]. Sedentary work at home may lead to a lack of hunger and desire for lunch, thus prolonging the starvation time inadvertently. However, postprandial blood glucose is known to increase after long starvation [12]. Therefore, in this study, we aimed to examine if skipping or serving a small portion of meal for lunch could affect the postprandial glucose levels at dinner.

To investigate the effect of a high protein breakfast meal and skipping/small lunch on the postprandial glucose level during an entire day, we conducted the following four trials: high protein or normal breakfast with or without lunch, respectively, with continuous glucose monitoring (CGM) to record the blood glucose levels according to the different eating patterns.

## 2. Materials and Methods

### 2.1. Participants

Nineteen healthy young adults (6 men and 13 women) aged from 20 to 36 years were recruited between February and June 2022. All the participants met the following 6 criteria: (1) had not been diagnosed with diabetes or dyslipidemia and not using drugs or supplements related to metabolism; (2) not taking antipsychotics, sleep medications, or steroids that may affect the circadian rhythm or clock gene fluctuations; (3) not highly obese (body mass index [BMI] > 35 kg/m^2^) or with sleep apnea syndrome; (4) no smoking habit; (5) no heart pacemaker or metal objects in their body (except silver teeth); and (6) not having worked or traveled to places with time differences within 2 weeks prior to the pre-survey (including during the study period).

This study was conducted according to the guidelines of the Declaration of Helsinki and was approved by the ethics committees of Tokyo Institute of Technology (approval number: A21231). This human trial is registered at www.umin.ac.jp/ctr/ (registration number, UMIN000048221). Each participant provided a written consent form following detailed description of the total experiment.

All the participants were asked to complete a questionnaire on the dietary intake (Food Frequency Questionnaire [FFQ] [23]), physical activity, exercise, and health status, prior to the study. None of the study participants were trained athletes competing in any sporting event. Seven participants were excluded from the analysis because of lost to follow up (n = 2), non-compliance with protocol (n = 3), and error of measurement in the glucose equipment (n = 3). Finally, 12 participants (three men and nine women) were included in this analysis (Figure 1).

### 2.2. Baseline Anthropometric and Physical Activity Measurements

On the first day, the body weight and basal metabolism rate (BMR) of all the participants were measured by a digital balance (Inbody 270, Inbody Inc., Tokyo, Japan). The participants were asked to wear a triaxial accelerometer (Active style Pro HJA-750C, Omron Corp., Kyoto, Japan) during the experimental period daily except during bath and sleeping times to monitor the intensity of physical activities. The data of physical activities were reported as metabolic equivalents (METs), and ranged from 0 (lowest) to 8 (highest). Only the data wherein the wearing time of accelerometer was >600 min per day were considered as valid. The duration of daily moderate-to-vigorous physical activity (MVPA) was calculated as a weighted average of weekday and weekend activities (i.e., weekly MVPA = [average daily weekday MVPA × 5] + [average daily weekend MVPA × 2]), to estimate the weekly activity. All the recordings ≥3 METs were classified as MVPA. 

### 2.3. Main Trials and Glucose Measurement

A randomized, crossover design was used in this study. All the participants were asked to participate in the following four trials in random order, in two weeks: (1) protein + skip (PS); (2) protein + lunch (PL); (3) normal + skip (NS); and (4) normal + lunch (NL). The first trial began at least 24 h after wearing the CGM system (FreeStyle Libre Pro, Abbott Laboratories, Chicago, IL, USA), for a 14-day CGM at 15 min intervals. During every trial, the mealtime was set at 8:00 for breakfast, 13:00 for lunch, and 18:00 for dinner. Each participant was asked to eat each meal within a 30 min period. Before the trial day, a pre-trial day’s dinner was also arranged as a test meal on 18:00 (Figure 2). From 18:00 on the pre-trial day till 24:00 on the trial day (in total, 30 h), the participants were allowed to eat only the test meal and water.

### 2.4. Test Meals

#### 2.4.1. High Protein Breakfast

Nutritional facts of the test meal were listed for the men (Table 1) and for the women (Table 2), respectively.

Men: salad chicken (FamilyMart, Shinjuku-ku, Tokyo, Japan), salad fish (Lawson, Shinjuku-ku, Tokyo, Japan), a cup of yogurt (DANONE Japan Co., Ltd.), 200 mL of milk protein (SAVAS cocoa, Meiji, Shijuku-ku, Tokyo, Japan), a protein bar chocoalmond taste (Morinaga, Shinjuku-ku, Tokyo, Japan), 25 g whey protein powder (Glico, Shinjuku-ku, Tokyo, Japan), and 5 g inulin powder (Fuji Nihon Seito, Chuo-ku, Tokyo, Japan) containing 653 kcal.

Women: salad chicken (FamilyMart, Japan), a cup of yogurt (DANONE Japan Co., Ltd., Shijuku-ku, Japan), 200 mL of protein (SAVAS cocoa, Meiji, Japan), a protein bar chocoalmond taste (Morinaga, Japan), 20 g whey protein powder (Glico, Japan), and 3 g inulin powder (Fuji, Japan) containing 536.7 kcal.

The inulin powder was added to adjust the meal to the same dietary fiber level compared to that of the normal breakfast.

#### 2.4.2. Normal Breakfast

Men: 1.75 calorie mate jelly apple (Otsuka Pharmaceutical, Tokyo, Japan) and 1.75 protein bar vanilla taste (Morinaga, Japan) containing 691.3 kcal.

Women: 1.4 calorie mate jelly apple (Otsuka Pharmaceutical, Tokyo, Japan) and 1.4 protein bar vanilla taste (Morinaga, Japan) containing 525.4 kcal.

#### 2.4.3. Lunch

On the skipped days, no lunch was served. On the lunch days, they received 0.5 calorie mate jelly apple (Otsuka Pharmaceutical, Tokyo, Japan) and 0.5 protein bar vanilla (Morinaga, Japan) containing 197.5 kcal for the men and 0.4 containing 158 kcal for the women. The small amount of lunch was designed just to maintain but not to cover the effect of breakfast.

#### 2.4.4. Dinner before the Trial Day

Men: frozen beef stew (Kenkosansai, Kita-ku, Yokohama, Japan), 250 g rice (Hagoromo, Shijuku-ku, Tokyo, Japan), 200 mL milk (Morinaga, Japan), and a box of crisp choco (Nissin, Shinjuku-ku, Tokyo, Japan) containing 992.5 kcal.

Women: frozen beef stew (Kenkosansai, Japan), 200 g rice (Hagoromo, Japan), and 200 mL milk (Morinaga, Japan) containing 671 kcal.

#### 2.4.5. Dinner on the Trial Day

Men: frozen tofu gratin (Kenkosansai, Japan), 250 g rice (Hagoromo, Japan), 200 mL milk (Morinaga, Japan), and a box of crisp choco (Nissin, Japan) containing 971.5 kcal. 

Women: frozen tofu gratin (Kenkosansai, Japan), 200 g rice (Hagoromo, Japan), and 200 mL milk (Morinaga, Japan) containing 650 kcal.

### 2.5. Statistical Analysis

All the data are presented as mean ± standard error of mean (SEM). GraphPad Prism version 8.3.1 (GraphPad software, San Diego, California, USA) was used for all the statistical processing of the experimental data in this study. To evaluate the postprandial glucose level, the incremental area under the curve (iAUC) was calculated by adding the areas of the trapezoids above the fasting level between every two continuous time points. The normality and equality of the data were examined by the Shapiro–Wilk test and Bartlett’s test. For the contrast among the four groups, parametric data that passed the normality test were analyzed by two-way repeated ANOVA, and Sidak test was performed for the post hoc test. Non-parametric data were adjusted by FDR and analyzed by two-way repeated ANOVA. For the contrast between two groups, parametric data that passed the normality test were analyzed by paired t-test while the non-parametric data were analyzed by Wilcoxon matched-pairs signed rank test. Statistical significance was set at *p* < 0.05 (two-tailed).

## 3. Results

### 3.1. Physical Characteristics

The baseline age, height, body weight, BMI, BMR, daily energy intake, fat rate, and muscle mass were measured are shown in Table 3. 

In each trial day, there was no significant different between groups in the physical activity evaluated by step counts and MVPA (Table 4).

### 3.2. h Changes in Glucose Pattern

The glucose data were recorded using a sensor every 15 min. The average data for every hour was taken to plot the 24 h glucose curve (Figure 3). For example, ‘7′ on the horizontal axis refers to the average glucose level at 7:00, 7:15, 7:30, and 7:45; therefore, the 8:00 breakfast is shown by an arrow pointing to ‘7’.

### 3.3. h iAUC at Breakfast, Lunch, and Dinner

Considering the error caused by the different fasting glucose levels, only the area above the fasting glucose level for each participant’s AUC was considered, to evaluate the postprandial glucose response as iAUC. The fasting glucose level was determined as the glucose level at 7:45, just before breakfast. Therefore, 16 h (8:00~24:00) iAUC was calculated to evaluate the total glucose level on each trial day (Figure 4). The NL group showed significantly higher 16 h iAUC than PL group (*p* < 0.05). There was a small but non-significant tendency with NS > PS (*p* = 0.68).

By 3 h after breakfast, the iAUC of the NS group was significantly higher (*p* < 0.001) than that of the PS group, while that of the NL group was also significantly higher (*p* < 0.0001) than that of the PL group (Figure 5). There was no significant difference between the two normal or two protein groups. We showed that high protein breakfast led to much lower glucose response than normal breakfast. 

By 3 h after dinner, the iAUC of the NS group showed no difference compared to that of the PS group, while that of the NL group was significantly higher (*p* < 0.05) than that of the PL group (Figure 5). There was no significant difference between the two normal or two protein groups. We showed that high protein breakfast only suppressed the postprandial glucose after dinner when breaking the 10 h fasting between breakfast and dinner.

By 3 h after lunch, the iAUC of the normal groups showed no difference compared to the protein groups, with or without skipping of lunch (Figure 5). Considering the rather small lunch, with only about a quarter of breakfast or dinner, the iAUC at 1.5 h after lunch was calculated. By 1.5 h, the NL group showed significantly higher (*p* < 0.05) iAUC than the PL group, which suggests that high protein breakfast suppressed the postprandial glucose level after a small lunch.

### 3.4. There Was No Difference between Genders

Results of the postprandial iAUC were showed in Figure 6 by women and men separately. No significant difference was observed between genders (Table 5). There are similar patterns of postprandial iAUC in women (N = 9) and men (N = 3) (Figure 6) in comparison with iAUC observed as total participants (N = 12) (Figure 5).

## 4. Discussion

In this study, high protein breakfast significantly suppressed the postprandial glucose iAUC (*p* < 0.01 or *p* < 0.0001) after breakfast, which suggests a strong effect of controlling the postprandial glucose level after some meals. The 3 h iAUC at lunch did not show a significant difference between NS and PS or NL and PL groups. Since the lunch was designed as just a small meal to break the long fast between breakfast and dinner, the energy required at lunch was only about a third of that at breakfast and a fifth of that at dinner. Besides, from the glucose curve, we observed that the postprandial glucose level at dinner took about 6 h to return to the fasting level while that of lunch took about 3 h. Considering the above, the 1.5 h iAUC at lunch was calculated to obtain the postprandial glucose level at lunch, and the PL group showed significantly lower level than NL group. Therefore, high protein breakfast suppressed the postprandial glucose level after lunch, despite the small amount at lunch.

The postprandial glucose level at dinner was suppressed in the PL group compared to that in NL group while it was not suppressed in the PS group compared to that in NS. This finding suggests that high protein breakfast continued to suppress the postprandial glucose at dinner. However, without lunch, even if dinner was the second meal, the high protein breakfast did not suppress the postprandial glucose significantly. We consider that the 10 h fasting was so long that the lowered free fatty acid (FFA) and other changed factors had rebounded to the fasting level and had lost the potential to influence the second meal. However, if a small lunch were eaten in the middle of the fasting, their functions could have been maintained till dinner. Therefore, for people that usually skip their lunch during busy days, even if they eat more healthy breakfast, the function could be attenuated by the prolonged fasting, which suggests the importance of lunch in the daily glucose control. Although a small amount of lunch like snack did not help reduce the risk of postprandial hyperglycemia (NS vs. NL), the loss of this small lunch could make the efforts of eating breakfast go to waste. Additional evidence based on the blood contents is needed to explain the difference between shorter and longer fasting.

In previous studies, several hypotheses were raised to explain the second meal effect, mainly including the function of FFA, insulin, and physical gastric emptying [24]. Along with different researches, controversial conclusions were also reached due to differences in participant characteristics and meal composition, size, and administration methods [25]. The role of FFA is universally acknowledged; in most studies, fasting is reported to increase FFA concentration, which increases the postprandial glucose response at meals after a prolonged fasting by increasing insulin resistance, hepatic glucose [26,27], and reducing glucose oxidation [28]. A meal to break the fasting state could reduce the concentration of FFA immediately, which could suppress the postprandial glucose by increasing muscle glycogen storage or relieving insulin resistance [27,29,30]. High protein breakfast has been proved to have a stronger effect on reducing the FFA level raised by fasting [31]. Aside from the effect of releasing the insulin resistance caused by FFA, high protein food has been proved to have a direct effect on promoting insulin secretion [32] or decreasing hepatic insulin clearance [33]. Besides, there is also a report showing that pre-load of protein before a meal could induce the release of peptides like glucagon-like peptide-1 (GLP-1), gastric inhibitory peptide (GIP), and cholecystokinin (CCK) in the small intestine, which serve for slowing gastric emptying and promoting insulin secretion [24]. On the other hand, protein rich breakfast means a low ratio of carbohydrate and lipid; therefore, such low ratio of carbohydrate and/or lipid may produce a second meal effect. In future, we will examine besides the protein rich meal, whether carbohydrate and lipid rich breakfasts also provide the second meal effect.

Activities in general, lowers the blood glucose level by intake of glucose into the skeletal muscle. In the present experiment, there were no significant stepping activity and MVPA differences among the four groups; however, the skipping lunch groups (NS and PS) showed lower values compared with the small lunch groups (NL and PL). As glucose levels at dinner was NL > NS and PS > PL, physical activity itself may not be involved in the glucose levels in the current experiments. In this experiment, high protein breakfast in the lunch groups may suppress the postpran dial glucose level at lunch, and subsequently at dinner, via the mechanisms described above. However, for the skipping lunch groups, with the 10 h long fasting period, no matter how high the protein or normal breakfast consumed, the FFA had risen to a much higher level, which caused a high insulin resistance so that the effect of the high protein breakfast was canceled or attenuated. On the other hand, the small lunch maintained the FFA level and served to maintain the effect of the high protein breakfast.

Although numbers of participants were small (3 male and 9 female), there were no clear differences of the postprandial iAUC in both genders, suggesting that a similar effect is expected in women and men. In the current experiments, approximate ratio is 40%, 10% and 50% of daily energy for breakfast, lunch and dinner, respectively. However, in future experiments, setting up a proper amount of lunch that is close to the daily routine (30-30-40, 30-20-50 or 20-20-60, etc. for breakfast-lunch-dinner) will be more ideal and the conclusion will be more convincing too at the same time.

## 5. Conclusions

When taking lunch, high protein breakfast could suppress postprandial glucose level after breakfast, lunch, and dinner compared with normal breakfast. However, when skipping lunch, high protein breakfast did not suppress the postprandial glucose level at dinner. Therefore, skipping lunch could attenuate the effect of a high protein breakfast which suppresses the post-dinner glucose.

## Figures and Tables

**Figure 1 nutrients-15-00085-f001:**
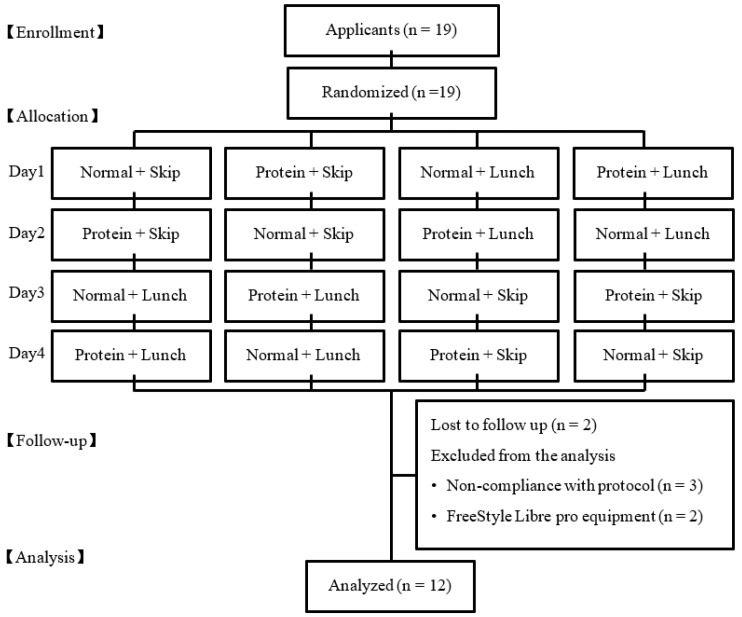
The flow diagram of the participants through the trials.

**Figure 2 nutrients-15-00085-f002:**
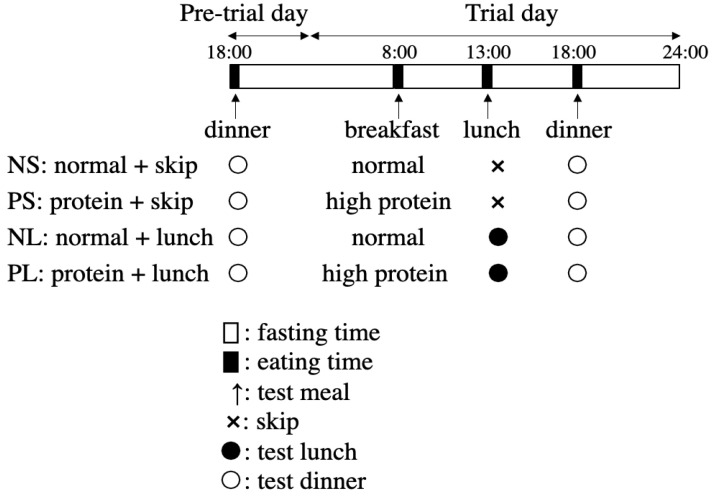
Schedule of the four main trials.

**Figure 3 nutrients-15-00085-f003:**
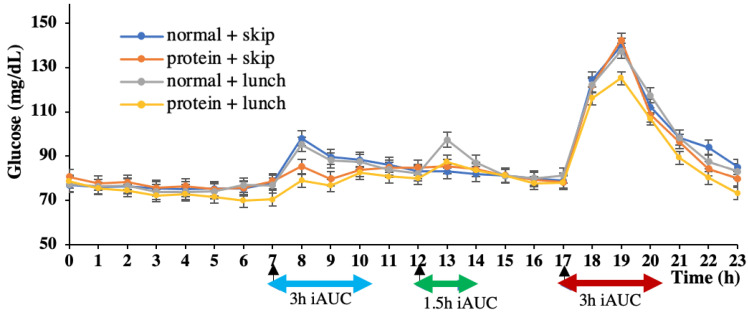
24 h glucose concentration pattern. Data are shown as mean ± SEM; iAUC, incremental area under the curve.

**Figure 4 nutrients-15-00085-f004:**
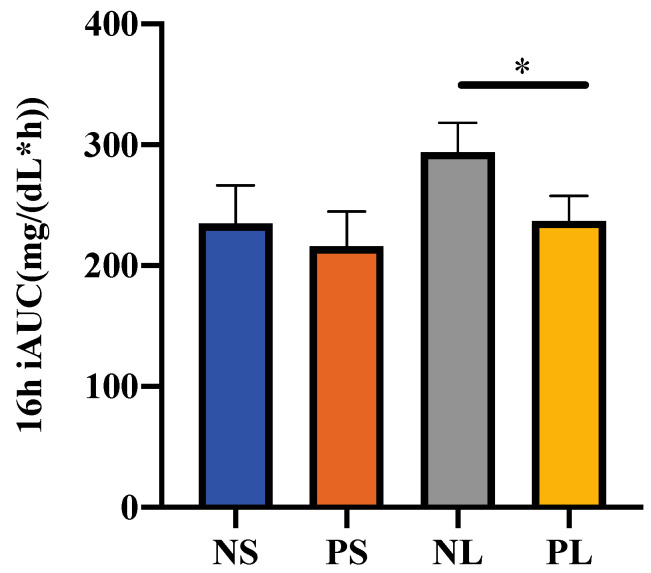
16 h iAUC of the postprandial glucose concentration. Data are shown as mean ± SEM. * *p* < 0.05, two-way repeated ANOVA, Sidak; iAUC, incremental area under the curve.

**Figure 5 nutrients-15-00085-f005:**
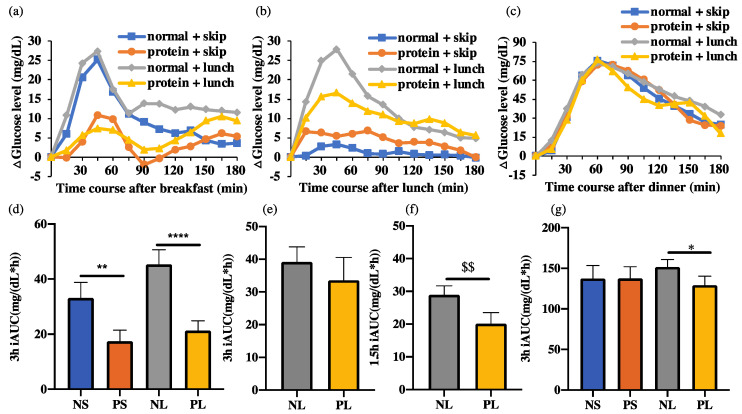
ΔGlucose levels 3 h after each meal and iAUC. Data are shown as mean ± SEM; iAUC, incremental area under the curve. NS, normal + skip; PS, protein + skip; NL, normal + lunch; PL, protein + lunch. (**a**) ΔGlucose levels 3 h after breakfast. (**b**) ΔGlucose levels 3 h after lunch. (**c**) ΔGlucose levels 3 h after dinner. (**d**) 3 h iAUC at breakfast. (**e**) 3 h iAUC at lunch. (**f**) 1.5 h iAUC at lunch. $$ *p* < 0.01, Wilcoxon matched-pairs signed rank test (**g**) 3 h iAUC at dinner. * *p* < 0.05, ** *p* < 0.01, **** *p* < 0.0001, two-way repeated ANOVA, Sidak.

**Figure 6 nutrients-15-00085-f006:**
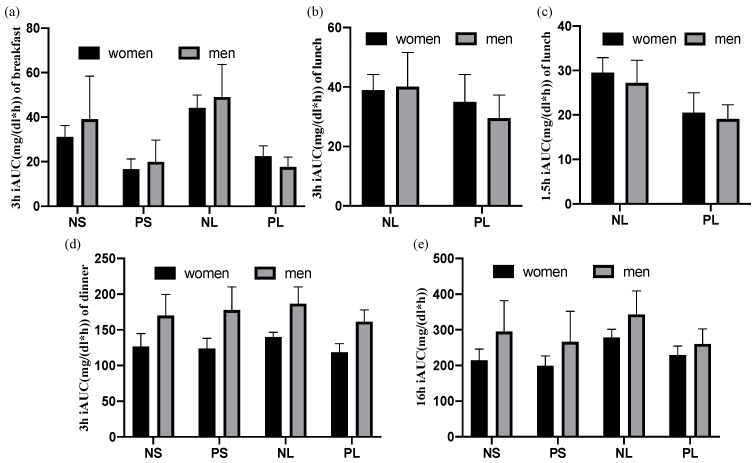
Postprandial iAUC of women and men separately. Data are shown as mean ± SEM. NS: normal + skip. PS: protein + skip. NL: normal + lunch. PL: protein + lunch. (**a**) 3 h after breakfast, (**b**) 3 h after lunch, (**c**) 1.5 h after lunch, (**d**) 3 h after dinner, (**e**) 16 h after breakfast.

**Table 1 nutrients-15-00085-t001:** Nutritional facts of the test meal for the men.

Nutrients	Dinner before the Trial Day	High Protein Breakfast	Normal Breakfast	Lunch	Dinner on the Trial Day
Energy (kcal)	992.5	653.0	691.3	197.5	971.5
Protein (g)	33.5 (13%)	98.3 (60%)	32.4 (18%)	9.3 (18%)	29.6 (12%)
Fat (g)	28.4 (26%)	12.1 (17%)	28.9 (36%)	8.3 (36%)	29.0 (27%)
Total Carbohydrate (g)	150.8 (61%)	37.4 (23%)	81.0 (45%)	23.2 (45%)	148.0 (61%)
Sugar (g)	145.4	27.0	70.5	20.2	143.1
Dietary fiber (g)	5.4	10.4	10.5	3.0	4.9
Salt (g)	1.9	3.0	0.5~0.9	0.1~0.3	2.6

**Table 2 nutrients-15-00085-t002:** Nutritional facts of the test meal for the women.

Nutrients	Dinner before the Trial Day	High Protein Breakfast	Normal Breakfast	Lunch	Dinner on the Trial Day
Energy (kcal)	671.0	536.7	525.4	158.0	650.0
Protein (g)	29.1 (17%)	81.9 (60%)	25.9 (18%)	7.4 (18%)	25.2 (16%)
Fat (g)	16.9 (23%)	8.9 (15%)	23.1 (36%)	6.6 (36%)	17.5 (24%)
Total Carbohydrate (g)	100.5 (60%)	35.0 (26%)	64.8 (45%)	18.5 (45%)	97.7 (60%)
Sugar (g)	95.9	26.8	56.4	16.1	93.6
Dietary fiber (g)	4.6	8.2	8.4	2.4	4.1
Salt (g)	1.6	2.3	0.4~0.7	0.1~0.2	2.3

**Table 3 nutrients-15-00085-t003:** Participant characteristics at baseline.

Baseline	Average	SEM
Age (years)	22.9	1.2
Height (cm)	163.8	2.9
Body weight (kg)	52.9	2.6
BMI (kg/m^2^)	19.6	0.4
BMR	1255.9	58.4
Energy intake (kcal/day)	1810.2	154.0
Fat (%)	22.9	1.9
Muscle mass (kg)	22.4	1.7

SEM, standard error of the mean; BMI, body mass index; BMR, basal metabolic rate.

**Table 4 nutrients-15-00085-t004:** Participant level of physical activity according to the four trials.

Trials	NS	PS	NL	PL
Step count (steps/day)	5632.8 ± 1061.6	6507.8 ± 1338.5	6745.5 ± 1119.6	7214.3 ± 1874.4
MVPA (min/day)	64.9 ± 13.9	68.3 ± 15.7	74.5 ± 12.8	73.8 ± 15.9

Data are shown as mean ± SEM. NS, normal + skip; PS, protein + skip; NL, normal + lunch; PL, protein + lunch; MVPA: moderate-to-vigorous physical activity.

**Table 5 nutrients-15-00085-t005:** Postprandial iAUC of different groups and genders. All data were presented as mean ± SEM (mg/(dL·h)). *p* value: two-way repeated ANOVA, FDR.

Group		Average (*n* = 12)	Women (*n* = 9)	Men (*n* = 3)	FDR-Adjusted *p* Value
3 h after breakfast	NS	33.16 ± 5.67	31.17 ± 5.09	39.13 ± 19.29	0.7242
	PS	17.44 ± 4.02	16.61 ± 4.61	19.92 ± 9.85	0.7815
	NL	45.38 ± 5.31	44.15 ± 5.76	49.04 ± 14.57	0.7778
	PL	21.3 ± 3.57	22.53 ± 4.58	17.63 ± 4.49	0.4713
3 h after lunch	NL	39.2 ± 4.57	38.90 ± 5.21	40.08 ± 11.50	0.9349
	PL	33.57 ± 7.01	34.93 ± 9.17	29.50 ± 7.76	0.7076
1.5 h after lunch	NL	28.92 ± 2.75	29.50 ± 3.40	27.17 ± 5.15	0.7575
	PL	20.07 ± 3.41	20.50 ± 4.49	19.08 ± 3.21	0.8512
3 h after dinner	NS	137.38 ± 16.04	126.46 ± 18.42	170.13 ± 29.59	0.2834
	PS	137.52 ± 14.39	124.10 ± 14.21	177.79 ± 32.37	0.2317
	NL	151.69 ± 9.26	140.03 ± 6.58	186.67 ± 23.51	0.1784
	PL	129.18 ± 11.22	118.44 ± 12.24	161.38 ± 16.73	0.1008
16 h after breakfast	NS	234.98 ± 31.39	214.89 ± 31.35	295.25 ± 86.36	0.4563
	PS	216.32 ± 28.6	199.56 ± 27.44	266.63 ± 85.25	0.5197
	NL	294.28 ± 23.95	278.02 ± 23.64	343.09 ± 65.79	0.4318
	PL	237.19 ± 20.8	229.31 ± 24.90	260.83 ± 41.29	0.5526

## Data Availability

Data will be sent upon request from the corresponding author. The data are not publicly available because of patent preparation.

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
