# Peer review of "Effect of a High Protein Diet at Breakfast on Postprandial Glucose Level at Dinner Time in Healthy Adults"

_nutrients, 2022, doi:10.3390/nu15010085_

Round 1
Reviewer 1 Report
Here, Xiao et al. reported an interesting effect of high-protein diet at breakfast on postprandial glucose at dinner in human subjects. They found that compared to normal breakfast, high-protein breakfast followed with lunch suppressed postprandial glucose level after breakfast, lunch as well as dinner. Furthermore, it was also shown that having no lunch reduced this effect of high-protein breakfast. The manuscript provided decent-quality dataset that supported the conclusion. However, there are several issues that need to be changed:
1. The title" second meal effect of a high protein diet at breakfast on post prandial glucose level at dinner time in healthy adults" sounds intriguing and confusing. I understand what the author meant, but it would be better to be replaced with "Effect of a high protein diet at breakfast on post prandial glucose level at dinner time in healthy adults".
2. The participants include 3 males and 9 females, in which female components were dominant here. In this case, it is very essential to understand whether there is difference in terms of post prandial glucose between the male and female participants. Additional analysis by comparing post prandial glucose between all male and female participants will be needed here.
3. Have lunch or not was a major factor that determined effect of high-protein breakfast onto the post prandial glucose level at dinner. However, it seemed an extremely light lunch was designed and given to all the participants during the tests, which was only about "a third of that at breakfast and a fifth of that at dinner". Even though it is totally acceptable to set up specific amount for lunch here, setting up a proper amount of lunch that is close to the daily routine (30-30-40, 30-20-50 or 20-20-60 etc. for breakfast-lunch-dinner) will be more ideal and the conclusion will be more convincing too at the same time.
Reviewer 2 Report
As is, I have no further comments. The science looks well performed. The graphs are very informative and the paper is well written.
